# Alterations in Essential Fatty Acids, Immunoglobulins (IgA, IgG, and IgM), and Enteric Methane Emission in Primiparous Sows Fed Hemp Seed Oil and Their Offspring Response

**DOI:** 10.3390/vetsci9070352

**Published:** 2022-07-11

**Authors:** Mihaela Hăbeanu, Nicoleta Aurelia Lefter, Anca Gheorghe, Mariana Ropota, Smaranda Mariana Toma, Gina Cecilia Pistol, Ioan Surdu, Mihaela Dumitru

**Affiliations:** 1National Research Development Institute for Biology and Animal Nutrition, Balotesti, 077015 Ilfov, Romania; ciuca_nicoleta@yahoo.com (N.A.L.); anca.gheorghe@ibna.ro (A.G.); mariana.ropota@ibna.ro (M.R.); smaranda.pop@ibna.ro (S.M.T.); gina.pistol@ibna.ro (G.C.P.); mihaela.dumitru22@yahoo.com (M.D.); 2Mountain Economy Centre (CE-MONT), Romanian Academy “Costin C. Kiritescu” National Institute of Economic Researches, 050711 Bucharest, Romania; surdu.ioan@gmail.com or

**Keywords:** hemp oils, fatty acids, immunoglobulins, E-CH_4_, diarrhoea, sows, piglets

## Abstract

**Simple Summary:**

One of the main problems in the sows’ farrowing sector is related to the health of their piglets and environmental impact as well. Hemp seed oil, characterised by an excellent polyunsaturated fatty acid profile and a favourable n-6:n-3 ratio, can contribute positively to enriching sow diets in essential fatty acids that are beneficial to health. Our study highlights that a higher n-3 fatty acids intake by sows fed diet-based hemp seed oil determines a fluctuation in the plasma concentration of major sows and their offspring’s immune markers (IgA, IgG, IgM). The suckling piglets receive significant Ig and essential fatty acids via maternal secretion. A milk intake with an ameliorated n-3 polyunsaturated fatty acid composition reduces the diarrhoea frequency of piglets. Enteric methane, known as a greenhouse gas indicator, can be impacted positively as well. New potential predictors for enteric methane, such as feed intake, n-6:n-3 ratio, linolenic fatty acids, and lean meat can be considered. Throughout the lactation period, milk fatty acids and plasma immunoglobulins fluctuate significantly.

**Abstract:**

This study shows the effects of dietary hemp seed oil on the milk composition, blood immunoglobulins (Ig), and enteric methane (E-CH_4_) production of primiparous sows, and their offspring’s response at three time points. A bifactorial experiment was conducted for 21 days (d) on 18 primiparous sows (195 ± 3 days old). The sows were fed two diets: (i) a control diet (SO) based on soybean oil (1.6%), with an 18.82 n-6:n-3 polyunsaturated fatty acids (PUFA) ratio; (ii) an experimental diet (HO) based on hemp seed oil (1.6%), with a 9.14 n-6:n-3 PUFA ratio. The milk contained an elevated level of linoleic acids (LA), n-3 FA, and especially alpha-linolenic acids (ALA), while the n-6:n-3 ratio declined using hemp oil. The Ig concentration was higher in colostrum than in milk. In the first few hours, the IgG in the plasma of piglets was more than double that of maternal plasma IgG (+2.39 times). A period effect (*p* < 0.0001) for IgG concentration in the plasma of piglets was recorded (40% at 10 d, respectively 73% lower at 21 d than 12 h after parturition). However, the sow diet did not affect Ig (*p* > 0.05). The frequency of diarrhoea declined after about 7 d. The value of the rate of diarrhoea was 6.2% lower in the PHO group. We found a 4.5% decline in E-CH_4_ in the HO group. Applying multiple linear regression, feed intake, n-6:n-3 ratio, ALA, and lean meat were potential indicators in estimating E-CH_4_. In conclusion, sow dietary hemp seed oil increased lean meat %, milk EFA, and milk IgM. Significant changes in the other dependent variables of interest (body weight, plasma Igs in sows and offspring, E-CH_4_ production) were not recorded. There was reduced diarrhoea which shows that EFA could play a therapeutic role in the incidence of diarrhoea and in lowering of E-CH_4_ emission in sows and progeny. All dependent variables were significantly altered at different time points, except for fat concentration in milk and sow plasma IgG.

## 1. Introduction

Hemp is a variety of *Cannabis sativa*, commonly cultivated in temperate climates including Romania. Oil extracted from hemp seed is a by-product characterised by an excellent profile of polyunsaturated fatty acids (PUFAs) and a good ratio of n-6:n-3 PUFA [1]. The n-6 and n-3 PUFAs are not interconvertible and frequently have opposing functions. The linoleic acids (LAs), which belong to the n-6 PUFAs, and alpha-linolenic acids (ALAs), which are n-3 PUFAs, cannot be synthesised by animals and are considered essential fatty acids (EFAs). The diet must provide these EFAs as a starting point for synthesizing other unsaturated FAs [2]. Maintaining a healthy balance of n-6 to n-3 PUFAs is crucial for animal health and good development [3]. The PUFAs play an essential role in immunomodulation and lipid metabolism, being indispensable for developing the foetal immune system and inflammatory response and protecting against some diseases as well [2,4]. In previous studies aimed to increase litter performance and positively influence their health, specific attention was given to the colostrum and milk FA composition and the biological responses of offspring. Dietary PUFA composition might impact colostrum and milk immunoglobulin (Ig) components [5]. Piglets at birth have little development of the immune system and only trace Ig. The piglet’s passive immunity is acquired naturally through the consumption of colostrum and milk [6]. The ability of the pig to resist disease requires a well-developed immune system [4,7,8,9,10]. After birth, by passing through a microbe-rich environment, piglets are sensitive to gastrointestinal disorders [11]. The first sign of digestive dysfunction is diarrhoea [12,13], known as a common cause of morbidity and mortality. In the last several decades, antibiotics, which inhibit or kill bacteria in animal bodies, have been the most widely known and utilised antimicrobials for controlling infectious diarrhoea [14]. However, the overuse of antibiotics can damage intestinal health and develop a high level of microbial resistance. Alternative options were suggested when antibiotics were outlawed in 2006, including nutritional interventions based on bioactive compounds from certain feedstuffs.

On the other hand, several approaches have proven effective in reducing enteric methane (E-CH_4_) emissions while increasing animal productivity. However, the most commonly targeted species have been ruminants. Pigs also produce CH_4_ when digesting, although the amount is much lower than ruminants [15]. A minor and inconsequential impact on CH_4_ production was obtained by Jørgensen et al. [16] using either rapeseed oil or fish oil in growing pigs and Theil et al. [17] by using animal fat in adult sows. However, neither of these trials was expressly planned to examine the influence on CH_4_. The terminal ileum metabolises 94% of fat [18]. As a result, only small amounts of dietary FAs reach the large intestine, where they have the ability to reduce CH_4_ synthesis. Current approaches have suggested that high-oil feedstuffs and their by-products containing PUFAs in a high concentration could potentially diminish E-CH_4_ generation in ruminants [19,20] and monogastric animals [21]. Farm-gate emissions have been dominated by livestock production processes such as enteric fermentation and manure deposition on pastures, which have produced three billion tonnes of CO_2_ equivalent (eq.) [22]. According to EEA [23], the CH_4_ emission from enteric fermentation was 192,227 kilotonnes of CO_2_ eq. in EU-28, and Romania recorded the production of 10,682 kilotonnes of CO_2_ eq. (data refer to 2010). FAO [15] developed the model proposed as the Global Livestock Environmental Assessment Model (GLEAM) and provided the GHG incidences (expressed in CO_2_ eq.): beef cattle contribute 91% of E-CH_4_, dairy cattle 85%, buffaloes 91%, sheep 93%, and goats 93%, while pigs recorded about 11% of livestock total enteric fermentation and manure management [24].

The FAs commonly found in the colostrum and milk of sows (C14:0, C16:0, C16:1, C18:0, C18:1, C18:2) are heavily influenced by dietary FA composition [25] as quoted by Lauridsen [26]. We developed the hypothesis that modifying the FA composition of milk from sows by including hemp seed oil in the diet would positively impact the response of their offspring in terms of immunological and gut functionality because of the role of FAs in early piglet feeding in connection to gut health. Furthermore, a lower n-6:n-3PUFA ratio could modify the production of E-CH_4_, starting with the statement that lipid could impact GHG_s_ emissions.

Thus, the objective of this study was to investigate the effects of adding hemp seed oil into the lactation diets of primiparous sows on the milk composition, blood Ig, and enteric methane (E-CH_4_) production of sows and their offspring’s response at three different time points after farrowing.

## 2. Materials and Methods

### 2.1. Ethical Procedure

The biological trial was conducted at the farrowing unit of the experimental station of INCDBNA Balotesti. The farm’s hygiene status followed strict biosecurity protocols, according to Law 199/2018, Romania, regarding the protection of animals used for scientific purposes. The care and use of animals were performed and approved by the Ethical Committee (no. 52/2021).

### 2.2. Animal and Housing

A bifactorial experiment (2 × 3) was performed for 21 days (weaning day) on 18 healthy primiparous sows Topigs-40 hybrid [♀ Large White × Hybrid (Large White × Pietrain) × ♂ Talent, mainly Duroc], 195 ± 3 days old, 140 ± 5 kg weight, in the autumn season (Appendix A). Sows were moved to the farrowing accommodation on day 100 of gestation. The microclimate factors were set electronically (average 24 °C, 64.5 percent rH). The sows were housed in cement-floored pens with a simple structure (vertical iron bars) and a total cage size of 2.60 × 1.80 × 0.80 m, providing an enclosed heated area for the piglets. The animals were ear-tagged and distributed into two groups (9 sows per group, each animal being considered a replicate). The pregnant sows were injected intramuscularly with 1 mL of prostaglandin (75 g active material/mL) to induce parturition simultaneously (±12 h). The sows were internally dewormed using ivermectin (1.5 mL/50 Kg) and vaccinated against *Escherichia coli* (2 mL/head).

A total of 244 piglets were born, of which 21 were stillborn, and 19 died throughout the trial. In this study, the offspring were standardised at ten piglets/sow (90/group, ~equal sex ratio per group). Piglets were maintained within the same litter, ear-tagged, and weighed individually at 12 h, 10 days, and 21 days after farrowing. Those with weights of less than 1 kg at farrowing were distributed to nurse sows, considered unviable, and eliminated from the testing group. The separation of piglets from their sows-mother was carried out at 21 days to evaluate the changes that the piglets underwent.

### 2.3. Treatments

The research station SCDA Secuieni, Romania, delivered the hemp seed, Jubileu variety, to the 2E-Prod SRL, Alexandria, Romania, where the oil was extracted.

During the last 15 days of gestation approximately, the sows were fed twice a day with about 2.5 kg of a classical cereal-soybean meal-based diet, formulated to contain 12.32 MJ ME/kg, 15.4% crude protein, 0.79% lysine, 0.50% standard ileal digestibility lysine (SID Lys), and 5.0% crude fibre content. About two days before farrowing, sows were given hot wet feed (water:wheat, 3:1 *wt*:*wt*). The sows were fed two diets, each with a distinct oil FAs composition: (i) the control diet (SO) based on soybean oil (1.6%), with an 18.82 n-6:n-3 PUFA ratio; (ii) the experimental diet (HO) based on hemp seed oil (1.6%), with a 9.14 n-6:n-3 PUFA ratio. The ingredients and diet composition are summarised in Table 1.

Throughout the experiment, sows were fed individually twice a day, and refusals were recorded. To get used to the consumption of solid feed, ten days after farrowing, the piglets started to receive a small amount of pelleted feed, which consisted of a classical diet based on corn (64.8%) and soybean meal (22%). To cover energy requirements, soybean oil was included in the diet (0.8%).

The amount of oil in the diet has been established to supplement energy needs without compromising other nutrient requirements relevant to the age and weight category. All diets contained crystalline amino acids (L-lysine and DL-methionine). The diets’ energy, protein, amino acids, and standardised ileal amino acids, calcium, and phosphorus content were similar between groups.

No antibiotic treatment was conducted on the animals throughout the experiment.

### 2.4. Measurements

Individual body weight (BW) was determined using an electronic scale (SWS International, Bucharest). Sows and their offspring were weighed at different time points after farrowing (12 h, 10 days, and 21 days). The feed intake and leftover amounts were recorded daily for sows and after ten days for piglets.

The Piglog105 version 3.1 portable equipment (SFK Technology A/S, DK-2730 Herlev, Denmark) was used to measure the back-fat thickness and muscle thickness, and lean meat content was calculated. Measurement to estimate the lean meat content of the animal was made at two predetermined anatomical points: between the third and fourth last lumbar vertebrae, 7 cm from midline; and between the third and fourth last rib, 7 cm from the midline. After entering our variable data (e.g., animal ID, pen location, age, and weight), we placed the ultrasonic sensor on the back of the animal at the correct measurement sites to estimate the lean meat content. The PC program “PigCom2” was used to transmit and read the data from the Piglog105.

### 2.5. Sampling

Colostrum samples (n = 9/treatment) were obtained within 12 h after farrowing, while milk samples were collected on days 10 and 21 of lactation. After the morning suckling, the sows were injected with 10 IU oxytocin (Veyx-Pharma, Schwarzenborn, Germany) in the auricular vein. The suckling piglets were separated from their mother for two hours to facilitate sampling. Colostrum and milk were collected manually from all functional mammary glands. The samples were stored at −80 °C until assay.

Blood samples (about 6 mL) were collected from sows first parity and their offspring from the jugular vein in heparinised tubes at different points (12 h, 10 days, and 21 days after farrowing). Serum samples were prepared by centrifugation for 5 min at 3500 rpm × g (Fuge D06, Neuation Technologies Pvt. Ltd., Gandhinagar, India).

### 2.6. Analyses

#### 2.6.1. Chemical Composition

The analyses were performed in duplicate. The gross chemical composition of the feed was determined using standardised methods according to Commission Regulation (EC) no. 152 (2009) [28]. Briefly, for crude protein, a semiautomatic classical Kjeldahl method was used (Auto 1030 Analyzer, Tecator Kjeltek) according to SR EN ISO 5983-2, 2009 [29]. According to SR ISO 6492, 2001, the fat was extracted using an adapted version of the classical procedure, which involved continuous solvent extraction followed by fat measurement using Soxhlet after the solvent was removed [29]. The cellulose was extracted using an intermediate filtering method according to European Commission (EC) Regulation no. 152 (2009) and the standard SR EN ISO 6865:2002 [21]. Van Soest extractions were used to determine the neutral and acid detergent fibre (NDF, ADF) according to SR EN ISO 16472:2006 and SR EN ISO 13906:2008. The Raw Fiber Extractor FIWE 6 (Velp Scientifica, Usmate, Italy) was used to conduct the analyses.

The tetrahydrocannabinol (Δ9THC) level of the hemp seed was determined according to Commission Delegated Regulation (EU) 2017/1155, Annex 3 [30] at the Central Laboratory for Wine Quality Control and Hygiene, Blaj, the only one accredited in Romania for THC analysis. A gas chromatograph fitted with a flame ionisation detector and a split/splitless injector, column 25 m long and 0.22 mm in diameter, impregnated with a 5% non-polar phenyl-methyl-siloxane phase was used.

Colostrum and milk samples were analysed for total solids, fat, protein, and lactose content by infrared spectrometry (ISO 9622:2013 IDF 141:2013) in the Laboratory of Physical-Chemical Analysis for Milk of the Romanian National Agency for Breeding and Reproduction in Animal Husbandry “Prof. Dr. G.K. Constantinescu” Balotesti (Ilfov county).

#### 2.6.2. Fatty Acids Composition in the Milk of Sows

For FA determination, we used a Perkin Elmer–Clarus 500 gas chromatograph (Boston, MA, USA), fitted with a flame ionization detector (FID) and capillary separation column with high polar stationary phase Agilent J&WGC Columns (Santa Clara, CA, USA), DB-23 with dimensions of 60 m × 0.250 mm × 0.25 μm, as previously described by Hăbeanu et al. [31].

#### 2.6.3. Ig Composition in the Colostrum and Milk of Sows

Immunoglobulin (Ig) subsets (M, G, and A) were measured by ELISA (Bethyl, Medist, Montgomery, TX, USA) in milk and colostrum centrifuged at 2700× *g*, 20 min at room temperature and then diluted in phosphate-buffered saline pH 7.0 (PBS, Dulbecco A; Oxoid Livingstone Ltd., London, UK), PBS with 1% BSA, and 0.05% Tween 20 as follows: 1:60,000 colostrum and 1:50,000 milk for IgM; 1:500,000 colostrum and 1:400,000 milk for IgG; and 1:100,000 colostrum and milk for IgA. According to the manufacturer’s instructions, goat anti-pig IgM, IgG, and IgA (1:20,000) and anti-pig streptavidin-conjugated horseradish peroxidase IgG (1:40,000) antibodies were used as primary and secondary antibodies. Absorbance was read at 450 nm using a microplate reader (Tecan Sunrise, Salzburg, Austria). Dilutions of recombinant swine reference serum were used as standards, and data were analysed with the linear segment of the generated standard curve. Results are expressed as mg Ig/mL of colostrum or milk.

#### 2.6.4. Ig Composition in Plasma of Sows before and after Farrowing and Their Offspring

The total plasma concentrations of immunoglobulin (Ig) subsets (G, A, and M) were measured by ELISA (Bethyl, Medist, Montgomery, TX, USA) in blood plasma after plasma dilution, as follows: 1:4000 (IgA), 1:120,000 (IgG), and 1:10,000 (IgM), according to the manufacturer’s instructions. Absorbance was read at 450 nm using a microplate reader (Tecan Sunrise, Salzburg, Austria). All samples included two duplicates for analyses. The values used in the statistical software represent the average of the duplicates.

### 2.7. Severity and Incidence of Diarrhoea of the Suckling Piglets

The suckling piglets were visually monitored twice a day to detect any piglets showing signs of diarrhoea. These health indicators were evaluated over the entire experimental period. A scoring system was used to indicate the presence and severity of diarrhoea as follows: one = hard faeces; two = soft-mild faeces; three = watery, mucous-like faeces [8]. When the average score exceeded three, the pigs were diagnosed with diarrhoea. The diarrhoea incidence of the piglets was estimated using the following formula:Frequency = [(total number of piglets with diarrhoea × days of diarrhoea)/(total number of piglets × days of the experiment)] × 100%(1)

### 2.8. E-CH_4_ Prediction

For E-CH_4_ evaluation, the model described by Hăbeanu et al. [21] was used. Briefly, the equation developed by Philippe and Nicks [32] is as follows:E-CH_4_ = 0.012 × dRes × DM intake (g CO_2_ eq·day^−1^)(2)
where dRes refer to digestible residues computed by INRA-AFZ (2004), as cited by Philippe and Nick [32]. The INCDBNA Balotesti database provided the theoretical digestibility coefficient. Starch and sugars were assumed to be 100% digestible.

### 2.9. Calculation

The metabolisable energy (ME) was calculated based on the feed composition and theoretical regression coefficients. The standard ileal digestibility (SID) of limiting amino acids was calculated using the feedstuff amino acids composition and the theoretical standardised ileal coefficient from CVB Feed 2021 (Table 1).

Colostrum intake (CI, g/piglet /day) was calculated after Theil’s [33] formula:CI, g = −106 + 2.26 WG + 200 BWb + 0.111 D − 1.414 WG/D + 0.0182 WG/BWb(3)
where WG = weight gain (g); D = duration of colostrum suckling (1440 min).

Weight gain (WG) was calculated using Devillers et al. [34].

Digestible organic matter (DOM) was calculated using the Le Goff and Noblet [35] formula:Sows: DOM = 1025 − 0.45 × NDF (g × kg^−1^ DM) − 2.03 × Ash (g × kg^−1^ DM)(4)
Piglets: DOM = 1035 − 0.72 × NDF (g × kg^−1^ DM) − 1.84 × Ash (g × kg^−1^ DM)(5)
where NDF means neutral detergent fibre.

The aggressiveness and frequency of diarrhoea were calculated, taking into account the days with diarrhoea (faeces score > 1).

### 2.10. Statistical Analyses

Statistical analyses were performed as a completely randomised bifactorial arrangement (2 × 3). The sows received two different treatments (SO and HO) over three time points (12 h after farrowing, 10 days, and 21 days after farrowing). The mean and pooled standard error of the mean (SEM) were used to express the findings. A two-way mixed repeated-measures analysis of variance (ANOVA in the IBM SPSS, 2011) was conducted, followed by a Bonferroni test for multiple comparisons to investigate the mean differences between groups. This approach reduces the variation caused by subject differences across many treatments. To avoid type I errors generated by sample size, Pillai’s trace was considered. The normality of data was checked by Shapiro–Wilks test. The interaction was investigated to test if the effect of one factor (diet) on the dependent variables was altered by the effect of another factor (time point of sampling). Each sow was considered an experimental unit and pen for the piglet. The piglets with missing data were eliminated from the statistical analyses. At *p* ≤ 0.05, the impact was declared to be statistically significant, and highly significant if 0.0001 < *p* < 0.01. Due to their insignificance (*p* > 0.05), the effect of replicates was removed from the study. We used the Pearson or nonparametric Spearman correlation to assess the bivariate association. Regression analyses were used to assess the strength of the relationship with predicted E-CH_4_. Multiple linear regression analyses were used to identify potential predictors and to evaluate the strength of the relationship for total E-CH_4_ production with some variable input.

## 3. Results

### 3.1. Dietary Oils Composition

The research station SCDA Secuieni, Romania, delivered the hemp seed, *Jubileu* variety, to the SC 2E-Prod SRL, Alexandria, Romania, where the oil was extracted. The hemp seed oil THC concentration was 0.014%. As shown in Table 2, the FAs composition of hemp seed oil vs. soybean oil revealed a higher value of ALA (1.84 times higher than that in the soybean oil, *p* < 0.0001), which was equal to the total amount of n-3 PUFAs. Although the n-6 PUFA values were almost similar among the two types of oil, the increased level of n-3 PUFAs led to a decrease of about 47% in the n-6:n-3 ratio (*p* < 0.0001).

### 3.2. Growth Parameters and Carcass Traits

#### 3.2.1. Growth Parameters and Carcass Traits of Primiparous Sows

The results with regard to growth parameters are presented in Figure 1a,b. There was a highly significant drop (−3.86% BW and −2.75% MBW ^0.75^, *p* < 0.001) at different time intervals.

The carcass characteristics (Figure 2a) results revealed a highly significant difference between groups for back-fat (BF) thickness and lean meat. In addition, while BF thickness declined up to 21 days (<20.15%), the lean meat increased (4.3%, *p* < 0.01, Figure 2b). The back-fat thickness negatively correlated with the muscle thickness (r = −89, *p* < 0.0001) and the lean meat (r = −94, *p* < 0.0001). Repeated measurements analysis revealed a highly significant interaction between diet and time for carcass traits determined in live animals (*p* < 0.01). Primiparous sows’ feed, fat, DM, and ME intake increased linearly until 21 days (weaning period, plus about 40%, from 3.56 on average in the first days rising to 5.96 at weaning, *p* < 0.0001), irrespective of diet (data presented in Appendix A).

#### 3.2.2. Growth Parameters of Piglets

Appendix A presents the ADFI, DMI, and colostrum intake of suckling piglets. The piglets’ BW and WG are shown in Appendix A.

### 3.3. Milk Composition of Lactating Primiparous Sows

#### 3.3.1. Effect of Diet and Time Point on Chemical and FAs Composition on Milk in Lactating Sows

*Milk chemical composition*. Except for the fat level, the other constituents, such as total solids and proteins, decreased up to 21 days (<28.14% and 65.36%, respectively, *p* < 0.0001), while lactose increased highly significantly (34.3% higher at 21 days compared to 12 h after farrowing, Table 3).

*Milk fatty acids composition*. The fatty acids composition of the first parity sows milk-fed a classical diet (SO) or an enriched n-3 PUFA diet (HO) at 12 h, 10 days, and 21 days after farrowing are presented in Table 4.

In the present study, the ⅀ PUFA, ⅀ n-3, ⅀ n-6, ALA, and LA concentration measured in milk was higher (*p* < 0.05) when 1.6% hemp seed oil was added to the diet. The ⅀ MUFA recorded a lower concentration (*p* = 0.004) in the HO group. The ratio of n-6:n-3 was 37.7% lower in the HO diet (*p* < 0.01).

Although the mean level of ⅀ PUFA was 22% higher in colostrum (12 h after farrowing) than ⅀ SAT and close to ⅀ MUFA, the milk concentration of ⅀ PUFA was lower than ⅀ MUFA and ⅀ SAT. We noticed an increase in ⅀ SAT at 10 and 21 days after farrowing (*p* < 0.0001). On the contrary, we noticed a significant decrease in ⅀ PUFA between 12 h and 21 days after farrowing (<38.89%). Although up to 10 days, the ⅀ PUFA, ⅀ n-3, ⅀ n-6, ALA, and LA concentrations decreased, the values started to record a slight increase up to 21 days (*p* < 0.01).

The ratio of n-6:n-3 was also altered by time due to the changes in n-6 and n-3 PUFAs (*p* < 0.0001). Furthermore, except for ⅀ SAT, a strong diet × time interaction was found (*p* < 0.05).

#### 3.3.2. Effects of Diets and Time Points on Igs Composition of Milk in Lactating Sows

IgM increased by 40.4% in milk from HO-fed sows (*p* < 0.0001, Table 5). However, the most predominant Ig in milk was IgG (>4.18 times higher than IgA and 6.51 times higher than IgM), while IgM had the lowest concentration (average 2.94 mg/mL).

The concentration of Ig changed during the stage of lactation. Thus, a higher level was recorded in colostrum than in milk, whatever the type of Ig. The post hoc test showed a pronounced decrease by 5.11 times for IgA, 11.64 times for IgG, and 3.59 times for IgM at T2 (10 days after farrowing) vs. T1 (12 h after farrowing). The decrease was maintained up to 21 days for IgG (*p* < 0.0001), while IgA and IgM increased slightly. There was a statistically significant interaction between diet and time for IgM (*p* = 0.012).

#### 3.3.3. Effects of Diets and Time Point on Igs Composition of the Plasma in Lactating Sows

Table 6 presents the Ig composition of the plasma in sows.

In this study, we observed a higher concentration of IgM in plasma than in milk (>55%) and a lower concentration of IgA (<45%). Except for IgG, the repeated measurements analysis showed both IgA and IgM were significantly affected by time. Thus, at 10 days after farrowing, a difference (<72%, *p* = 0.02) was recorded in IgA concentration. After 10 days up to 21 days, the IgA level recorded an increase (>32%, *p* < 0.0001). On the contrary, IgM increased up to 21 days (>4.2% at 10 days and 23% at 21 days vs. 12 h after farrowing, *p* < 0.0001).

#### 3.3.4. Correlation between Variable Ig of Milk and Plasma from the Sows and EFAs Composition

The Spearman coefficients of correlation among certain parameters are shown in Table 7.

The results presented in Table 7 show a strong Spearman correlation (*p* < 0.01) between sows’ milk IgA and ⅀ MUFA, ⅀ PUFA, LA, ⅀ n-6, ⅀ SAT, and the n-6:n-3 ratio. Similarly, milk IgG was negatively correlated with ⅀ SAT, ⅀ PUFA, LA, ⅀ n-6, and the n-6:n-3 ratio. IgM was strongly correlated with ⅀ MUFA, ⅀ PUFA, LA, and the n-6:n-3 ratio.

A greater relationship was found between plasma Ig and milk FA composition as follows: IgA was highly significantly correlated with ⅀ SAT, and negatively correlated with ⅀ PUFA, ⅀ n-6, LA, and the n-6:n-3 ratio; IgG did not record a significant relationship; IgM was highly negatively significantly associated with ⅀ SAT, ⅀ PUFA, ⅀ n-6, and LA.

#### 3.3.5. Ig Composition of the Plasma in Suckling Piglets

IgA showed the most important increase (>6.9% in the PHO group vs. the PSO group) (Table 8). Greater significant alterations were observed in Igs up to 21 days. The most pronounced concentration was noticed for IgG in the plasma of piglets at 12 h after farrowing (maximum value 57 mg/mL and minimum 38 mg/mL). Thus, at 10 days, the IgG concentration was 35% lower than on the first day (*p* < 0.0001), and the declining trend continued for up to 21 days, the average value being 73% lower compared to the first day after farrowing (*p* < 0.0001). A linear decrease was observed for plasma IgA concentration (<1.83 times at 10 days and < 3.98 times at 21 days after farrowing compared to the first day). IgM declined by 51% for up to 10 days (*p* < 0.01), and after that increased slightly compared to day 10 (>26.9%, *p* < 0.01).

#### 3.3.6. Correlation between Dependent Variables in Plasma Ig in Piglets and EFA in Milk

The Spearman coefficients of correlation among certain parameters are shown in Table 9. Milk EFAs were correlated with Ig in the plasma of piglets: IgA had a positive correlation with ⅀ PUFA, ⅀ n-6, LA, and n-6:n-3 and a negative relationship with ⅀ SAT; plasma IgG was correlated highly significantly with milk ⅀ SAT, ⅀ PUFA, ⅀ n-6, LA, and n-6:n-3 ratio. Similarly, IgM recorded a significant Spearman correlation with LA, n-6:n-3, ⅀ n-6, ⅀ PUFA, ⅀ MUFA, and ⅀ SAT.

### 3.4. Incidence Rate of Diarrhoea on Sows’ Offspring

Figure 3 presents the frequency of diarrhoea and the score of faeces consistency, giving us information about the severity of digestive disorders. The severity of diarrhoea was appreciated by assuring a score from 1 to 3 depending on the faecal consistency. The faecal consistency score had an average value of 1.64 (*p* > 0.05, Figure 1a). The addition of hemp seed oil to the sows’ diet resulted in a 6.4% reduction in diarrhoea in their offspring. The score of faeces consistency was higher in the PSO piglets’ group from sows fed a SO diet (16% higher, maximum value noted was 2.11; Figure 1a). The first signs of diarrhoea appeared 5 to 7 days after farrowing and lasted up to 17 days (Figure 1b).

### 3.5. E-CH_4_ Production

Total E-CH_4_ recorded a lower value in first parity sows fed a HO diet (<4.5%, *p* > 0.05, Table 10). On the contrary, a highly significant increase was recorded up to weaning time (−38%, *p* < 0.0001).

Our data supported the hypothesis of a nonzero association between E-CH_4_ and fat intake, n-6:n-3 ratio, ALA, and lean meat after applying a multiple linear regression model. These factors can be taken into account as potential E-CH_4_ predictors: fat intake (β coefficient = 0.99, R = 0.99, *p* < 0.0001), n-6:n-3 ratio, (β coefficient = −0.63, R = 0.69, *p* < 0.0001), ALA (β coefficient = 0.53, R = 0.53, *p* < 0.004), and lean meat (β coefficient = 0.43, R = 0.45, *p* < 0.02, Table 10). The independent variables mentioned above statistically significantly predict the dependent variable (E-CH_4_ production).

## 4. Discussion

This study showed that hemp seed oil administered to sows had the potential to increase lean meat %, milk EFA, and milk IgM. Nonetheless, significant changes in the other dependent variables of interest (body weight, plasma Igs in sows and offspring, E-CH_4_ production) in sows and their offspring were not identified. However, there was reduced diarrhoea and a lowering of E-CH_4_ emission in sows and progeny. On the other hand, significant alteration of all the dependent variables was observed during lactation, at different time points, except for fat concentration in milk and sow plasma IgG. Fat intake, lean meat, n-6:n-3 ratio, and ALA were found as potential predictors for E-CH_4_ emission.

Previously recorded data have shown that hemp seed has valuable nutritional characteristics [36], for use as feedstuff for dairy cows [37,38], lambs [39] and sheep milk production [40], poultry [41,42], and pigs [4]. These studies focused on hemp seed’s potential in monogastric feeding as a low-cost dietary component because it doesn’t necessitate irrigation, pesticides, or large amounts of synthetic fertilizers, and as a therapeutic feedstuff as suggested by the Borhade [43].

In our previous works [29,44,45], we have shown that a percentage of 5% of hemp seed in the diet can increase the n-3 PUFA level in the feed by around 53%, while the n-6:n-3 PUFA ratio can be decreased by 52% vs. in a classical diet. In this study, the dietary n-6:n-3 PUFA ratio was reduced by 48.6% due to an n-6 to n-3 level of 3.67 in hemp seed oil vs. 6.86 in soybean oil. In agreement with Lavery et al. [46], these changes in diet composition did not alter feed intake. Lavery et al. [46] mentioned that attaining high dietary unsaturated FAs by using salmon oil could have reduced feed, but peroxidation was minimal in the trial diets. Feed intake might be considered a major factor that impacts sows’ growth significantly during the lactation phase. A low appetite and hence low feed intake led to a reduced nutrient intake in the sows, despite a slight increase in final BW for the HO-fed group. Several studies examined various solutions for increasing feed intake to increase milk production and avoid reproduction problems which may lead to the premature culling of the sows [47,48,49]. However, due to limited intestinal capacity, the feed intake of the sows usually plateaus in late lactation [46].

On the other hand, increasing dietary energy density during late lactation could help improve sow energy intake in late lactation. Achieving this could increase milk production and subsequently piglet growth in late lactation. Furthermore, dietary oils usually increase the energy density and the palatability of diets [46]. Hăbeanu et al. [21] showed a positive correlation between feed intake and E-CH_4_, so a question remains, how we can find more suitable predictors?

It is well known that fat is inversely correlated with muscle thickness and lean meat. The current trial indicated an increase in muscle thickness up to weaning time, which led to a rise in lean meat proportion. This typically occurs during the lactation phase due to a large body tissue catabolism. However, the fat intake was similar between groups in the current study; the back-fat thickness recorded a dynamic decline due to diet. Our findings are consistent with those of Lavery et al. [46] who found that the type of oil or energy regimen had no effect on sow back-fat depth but that the time point did.

Feed for suckling piglets is frequently supplemented with LA and DHA to prevent PUFA deficiency [50]. We decided to use just a single classical diet to evaluate the impact of sow nutrition on offspring. After farrowing (AF), the offspring lose microbiological sterility by passing into a microbe-rich environment. The primary protection for the suckling piglets up to weaning is due to immunity assured by the milk, piglets being related to their mother–sows [31,51]. We can also presume that the percentage of piglets that were stillborn (8%) and died during the trial (about 10%) can be related to the physiological processes to which the piglets are exposed and their individual responses, while access to functional teats and birth weight could also be determinants.

However, in the current study, sow diets did not affect the growth parameters of piglets. In contrast to our findings, Lavery et al. [46] reported a high litter weight gain from sows fed diets containing salmon oil vs. soy oil.

Hăbeanu et al. [31] emphasised the impact of lipid supplementation on lipid profile in sows and piglet performance and their liveability. One of the reasons for this technique was the hypothesis that a high level of n-3 FA in the diets could lead to significant changes in the Ig concentration [3]. Whereas the transfer of Ig from the mother to infants is realised via the placenta [3], to the suckling piglets, the transfer is via the colostrum. Due to a higher positive correlation between FA composition in milk/colostrum of sows and Ig concentration in the plasma of their progeny, we could assume and confirm that, in the first days after farrowing, the colostrum intake has great importance. In previous research, Chen et al. [52] revealed that sows fed linseed oil had a higher concentration of ALA in milk than sows fed soybean oil. Moreover, Chen et al. [53] found that medium-chain FAs (mainly lauric acid) given to sows during lactation can reduce the incidence of diarrhoea in suckling piglets and increase the protein and main Igs’ concentration in colostrum.

Similarly, Vodolazska and Lauridsen [5], by adding hemp seed oil to the diet of sows in late pregnancy and the lactation phase, reported an enhancement in the milk composition of PUFA n-3, and piglets were able to convert these FAs received through sow milk consumption to C20:5 n-3 eicosapentaenoic (EPA) and C22:6 n-3 docosahexaenoic (DHA) acids. Furthermore, it was reported that hemp seed oil in the maternal diet increased the number of piglets born and decreased their mortality, affecting the concentration of Ig in the blood of the piglets. We hypothesise that we can enhance digestive health by altering the composition of milk. In this study, the total solids in milk were highest during the first 12 h and 10 days after farrowing and then declined, remaining at 19.7% at the final time point of the trial, consistent with Hurley’s [51] results. The fat and protein concentration had higher values than those reported by Hurley [51]. However, the tendency to decline up to 21 days was similar.

Concerning FAs composition, by hemp seed oil incorporation in the HO group, we found that both colostrum and milk contained an elevated level of n-3 FA, primarily ALA. However, the LA concentration increased while the n-6:n-3 ratio declined. We confirmed that the EFA composition in milk is impacted by the oil type used in sows’ feeding, made concrete by their FAs profile. We found lower EFA values in milk from primiparous sows than those found by Vodolaska and Lauridsen [5] in multiparous sows but higher than those found by Reis et al. [50] utilizing cow’s milk enhanced with n-3 and n-6 for sows’ feeding. Lavery et al. [46], by using salmon oil and different energy regimens, increased the proportions of the n-3 PUFAs, C20:5 n-3 eicosapentaenoic (EPA), and C22:6 n-3 docosahexaenoic (DHA), and decreased the proportions of the n-6 PUFA compared to soybean oil, whereas the ratio of total PUFA and n-6 PUFA was higher in milk at day 21 from sows fed soybean oil and the phased-energy regimen. Swiatkiewicz et al. [54] used two types of oils (coconut and rapeseed) which changed the milk FA composition differently. Thus, the coconut oil increased especially the lauric and myristic (C14:0) acids, while rapeseed oil increased LA and arachidic FAs, and eicosapentaenoic (EPA) and docosahehaenoic (DHA) acids. In our study, the effect of time point was more pronounced for PUFA, ⅀ n-6, and LA than ALA and ⅀ n-3. Evidence suggests that FAs influence the immune system [55,56,57], but the earliest review appeared in 1978 [58]. In 1993, Fritsche et al. [59] demonstrated that the FA profile and eicosanoid synthesis of piglet immune cells are affected by maternally administered fish oil, and n-6 and n-3 PUFAs can also modify the immune response [60]. One study by Vodolazska and Lauridsen [5] highlighted the transfer of FAs to offspring, the impact on piglet immunity, and the nutritional status of hemp seed oil included in the lactation multiparous sows’ diet. Pierzynowska et al. [4] evaluated the relationship between Igs and the fat absorption dependency of PUFA absorption on the presence of Igs in the blood of piglets after birth. The mechanisms of action and efficacy of the long-chain n-3 PUFAs found in seafood, particularly oily fish, and their effectiveness in inflammatory diseases have been studied extensively. On the contrary, we did not find literature data on the efficacy of hemp seed oil.

The Igs are considered the primary colostrum protein components. IgA, IgG, and IgM are the predominant Igs in pigs. During the entire lactation period, the Ig average values in our study were: 4.40 mg/mL IgA, 18.45 mg/mL IgG, and 2.94 mg/mL IgM. The IgG is the most important for the secondary immune response, while IgM is for the primary immune response.

IgA is essential for gastrointestinal health. In colostrum, we noticed a higher value for Ig than in milk. The addition of hemp seed oil increased the IgM concentration. The Igs fluctuated across time in different ways. Thus, the decline of IgA was approximately 80% at 10 days after farrowing compared with the values seen at farrowing, while at 21 days, we noticed a slight increase vs. 10 days after farrowing. We noticed a more pronounced decline for IgG (<91%) at 10 days compared to 12 h after farrowing. The decline continued up to 21 days. Our data concerning Igs in colostrum are comparable to those found by Yao et al. [3] in a diet with 9:1 n-6:n-3 PUFA obtained by dietary modulation of the ratio of n-6:n-3 PUFA using different corn oil and linseed oil percentages in a mixed oil supplement, while in milk our values for IgA and IgG were much higher.

Similarly, the IgA and IgM in the colostrum in sows were similar to those obtained by Hurley [51]. The same authors obtained a 28% lower concentration of IgG at 12 h after farrowing than our data. In a study by Corino et al. [61] quoted by Rossi et al. [62], sows fed with a supplement of 5 g/kg of CLA powder had higher IgG, IgA, and IgM concentrations in colostrum, as well as higher plasma IgG concentrations in nursing piglets.

The piglets ingest maternal Ig (IgA, IgG, and IgM isotypes) and other valuable nutrients by colostrum intake in the first days after birth. Piglets were fed only by maternal secretion for the first 10 days after birth, after which their diet was supplemented with a basal solid compound feed. The variation in the variables assayed could be attributed to their sows-mother feed composition. Surprisingly, the sows’ diets did not affect the litter Igs concentration in the plasma, although a slight increase was observed regardless of the kind of Ig.

Studies reporting the effects of n-3 PUFAs or n-6:n-3 PUFA on the suckling piglets have shown an inconsistency which has been repeatedly emphasised. Due to a higher positive correlation between sows’ milk and/ or colostrum FA composition and the plasma Ig concentration of their progeny, we could assume and confirm that the colostrum intake is vital during the first days AF. About 40% of the colostrum Ig comes from serum. After farrowing, within about 24 h, through colostrum sows’ secretion ingested by suckling piglets, IgG passes into the blood.

In contrast, after passing through blood, IgA circulates and reaches the respiratory epithelia and the intestinal epithelium. IgG was the most predominant Ig in milk or plasma from both sows and piglets. The highest levels were observed in the plasma of piglets in both groups. In the first hours after birth, the IgG in the plasma of the piglets was more than double that of the maternal plasma. Swiatkiewicz et al. [54] highlighted that the level of IgM and IgG in the plasma of piglets at 21 days depended on the type of oil received by the sows. Using coconut oil and rapeseed oil, higher contents of IgM and IgG were recorded in piglets from sows fed with coconut oil.

We cannot yet explain the mechanism of changing Ig concentration due to the dietary content of n-3 and n-6 FAs. However, the production of interleukins has been mentioned previously. The ratio between n-6:n-3 in sow’s diets in this research was higher than five; this value is specified to influence health parameters positively. Although, in our trial, the hemp seed oil was added in the same proportion (1.6%) as soybean oil, the ratio of n-6:n-3 was lower than in the SO diet. This allows us to stipulate that this ratio would have decreased linearly if the proportion of inclusion of hemp seed oil had been higher.

Diarrhoea in suckling and weaned piglets is one of their most predominant health problems. Gastrointestinal disorders result from the interaction between infective agents, host immunity, and management procedures, causing important economic losses to this sector. The gut is a vital immune organ. Non-pathogenic bacteria are helpful for digestive systems and are extremely important for balancing the microflora in the gastrointestinal tract. Colonization with pathogenic bacteria to piglets occurs rapidly after farrowing. Igs’ colostrum and milk composition protect piglets from potential damage produced by these pathogenic bacteria, especially *Escherichia coli*.

The period between birth up to about 7 days after weaning is critical for piglets exposed to multiple stressors associated with changes in the function and architecture of the gastrointestinal tract and enteric microbiota and immune response [8]. Piglets must rely on maternal protection (passive immunity) until their active immunity is developed. On the other hand, the digestive system of piglets suited for milk consumption is not entirely developed. However, it is essential to start consuming a supplement feed to develop their own immune and digestive systems. It is necessary to highlight that the diet must consist of easily digestible ingredients.

In this work, we obtained a significant decrease over time in the incidence of diarrhoea. We found that the frequency of diarrhoea was positively and significantly correlated with the days with diarrhoea (r = 0.78). About 7 days with diarrhoea were the maximum for one piglet, but the faeces were marked with two being soft-mild but not aqueous. The mean value of the rate of diarrhoea was 6.2% lower in the PHO group. Previous findings of Hăbeanu et al. [8] have shown lowering (better) faecal consistency scores by limiting protein in the nursery diet which is consistent with more recent results from Kroeskea et al. [63,64]. Oil derived from vegetable feedstuffs such as hemp seed, linseed, or algae is generally high in n-3 PUFA but weak in EPA and DHA, which have a higher bioactivity. One of the reasons why enzyme activities may be restricted, limiting evident impacts, is because of the desaturase process. More oxidative stress levels in the gut can compromise intestinal integrity [46]. In terms of anti-inflammatory properties, the n-3 PUFAs are more effective than n-6 PUFAs, but both FAs are sensitive to peroxidation, impacting gut health negatively. This could explain why the addition of hemp seed oil to the diet of sows has a diminished effect (*p* > 0.05).

The largest single source of CH_4_ in the EU-28 is the enteric fermentation of feed in the stomachs of livestock, especially ruminants. Compared to other productive animals, such as cattle, the CH_4_ emissions from pigs represent a very small percentage of all generated emissions. There are some investigations which have shown the inhibitory effect of ALA to CH_4_ in ruminants [65]. In contrast to SFA, which are less effective at reducing CH_4_ in cattle, C12:0, C18:3, and PUFA were mentioned by [65] as providing a considerable influence on decreasing CH_4_ emissions in ruminants. Our prior work discussed an effective method for decreasing E-CH_4_ generation in pigs by employing high fibre levels [21]. Our findings corroborate the idea that fat consumption influences E-CH_4_. However, E-CH_4_ production inhibition via manipulation of n-6:n-3 PUFA and n-3, particularly ALA, using hemp seed oil has not been reported yet. We intended to decrease the amount of CH_4_ produced by enteric fermentation while ensuring sustainable performance and health. Although the drop in sows and progeny was less significant, we found a 4.5% decline when using hemp seed oil in sows’ feeding. One possible explanation consists of the adverse effects of dietary EFA on microorganisms, but the relatively low dietary fibre content as a primary fermentation substrate may be sufficient to avoid a significant decline.

Our findings are in accordance with those of Nguyen et al., 2020 [66], who reported that using different n-6 to n-3 FAs ratios in diets based on corn–soybean meal seemed to have no effect on gas emission in growing pigs. We presume that the type of diet and the concentration of EFAs may influence how much E-CH_4_ is reduced. On the other hand, suckling piglets should not produce CH_4_ since sow’s milk does not contain fibre or polysaccharides. The solid feed contains fibre, which results in low CH_4_ emissions.

Applying multiple linear regression, four parameters could have a good potential in estimating E-CH_4_ concentrations (feed intake, n-6:n-3 PUFA, ALA, and lean meat). In another investigation [21], volatile fatty acids and microbiota were mentioned as factors that could impact E-CH_4_ emissions in growing pigs.

## 5. Conclusions

In summary, EFAs in milk may be modulated by adding hemp seed oil to sows’ diets. Based on our findings, the concentrations of n-6 and n-3 PUFA in milk are correlated, affect the Ig concentration in plasma, and reveal the mitigating potential of E-CH_4_. For a more pronounced effect, a higher level of oil rich in n-3 PUFA in the diet will be necessarily associated with a higher fibre level.

The Ig concentration in the plasma of both primiparous sows and their offspring fluctuated over time. In the lactation phase, the health of piglets depends on the structure of milk fat, including the days on which the milk is supplemented with solid food. Thus, this study confirms the assumptions that EFAs could play a therapeutic role in the incidence of diarrhoea, as piglets can convert this AF consumed by maternal milk. We point out some valuable indicators which can be included in equations for predicting E-CH_4_.

This nutritional strategy can be more effective, especially if an optimum level of dietary EFAs, especially n-3 PUFA, can be established. Additional in vivo studies are necessary to confirm these results.

## Figures and Tables

**Figure 1 vetsci-09-00352-f001:**
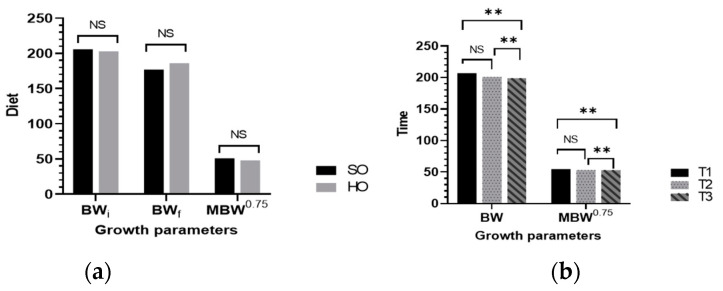
Mean of growth parameters of primiparous sows: (**a**) effect of diet (classical diet based on soybean oil, SO, or a rich n-3 FA and lower n-6:n-3 ratio diet based on hemp seed oil, HO) on BW and MBW; (**b**) effect of time (12 h, 10 days, and 21 days) on BW and MBW. Abbreviations: body weight, BW; metabolic body weight, MBW ^0.75^; Time: T1 = 12 h after farrowing, T2 = 10 days, T3 = 21 days; NS: nonsignificant effect; ** *p* < 0.01 highly significant difference between means.

**Figure 2 vetsci-09-00352-f002:**
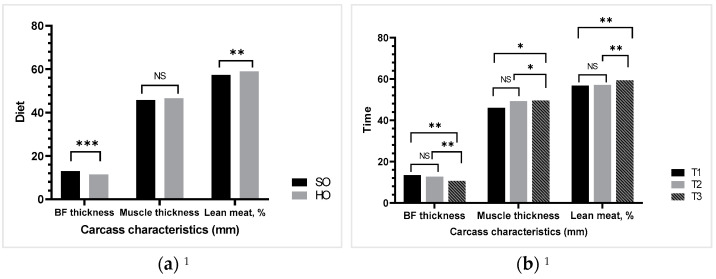
Mean of carcass characteristics of primiparous sows: (**a**) effect of diet (classical diet based on soybean oil, SO, or a rich n-3 FA and lower n-6:n-3 ratio diet based on hemp seed oil, HO) on BW and MBW; (**b**) effect of time (12 h, 10 days, and 21 days) on BW and MBW. ^1^ Piglog105 version 3.1 portable equipment (SFK Technology A/S, DK-2730 Herlev, Denmark) was used for measuring back-fat thickness and muscle thickness to calculate the lean meat content. Abbreviations: back fat, BF; Time: T1 = 12 h after farrowing, T2 = 10 days, T3 = 21 days; NS: nonsignificant effect; * *p* < 0.05 significant difference between means, ** *p* < 0.01 and *** *p* ≤ 0.001 highly significant difference between means.

**Figure 3 vetsci-09-00352-f003:**
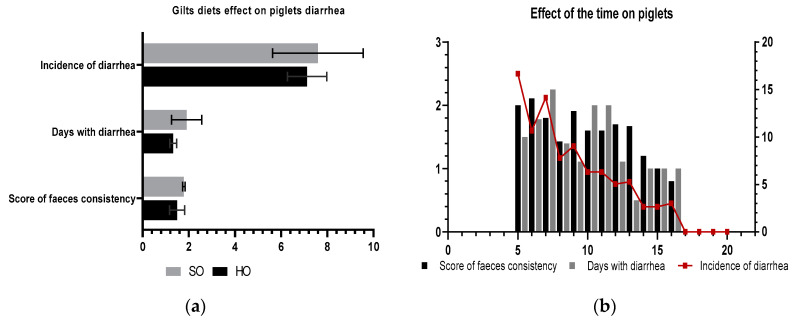
The diarrhoea severity and frequency trend: (**a**) effect of sows’ diets on piglets’ diarrhoea; (**b**) piglets’ diarrhoea over time. Every day piglets were examined visually. The score of faeces consistency applied: two for soft-mild diarrhoea; three for aqueous faeces, which means the presence of severe diarrhoea.

**Table 1 vetsci-09-00352-t001:** Ingredients and nutrient composition for Topigs-40 hybrid primiparous sows and their offspring’s diet.

Items (g × kg^−1^)	Lactating Sows Diets		Piglets Diet
	SO	HO	PSO and PHO
Corn	568.7	568.7	648.0
Rice bran	100.0	100.0	0
Soybean meal	180.0	180.0	220.0
Sunflower meal	100.0	100.0	0
Soybean oil	16.0	0	8.0
Hemp seed oil	0	16.0	0
Corn gluten feed	0	0	30.0
Milk replacer	0	0	50.0
DL-Methionine	0	0	0.9
L-Lysine	0.2	0.2	3.0
Calcium carbonate	17.5	17.5	14.6
Monocalcium phosphate	1.5	1.5	13.4
Salt	4.0	4.0	1.0
Choline premix	2.0	2.0	1.0
Vitamin-mineral premix ^‡^	10.0	10.0	10.0
Phytase (500 FTU kg·feed^−1^)	0.1	0.1	0.1
Analysed Composition (g·kg^−1^ as feed bases)			
Dry matter	879.1	879.1	891.0
Crude protein	189.4	189.4	189.0
Ether extract	51.4	51.4	36.4
Lysine	8.2	8.2	12.0
Met + Cys	6.9	6.9	7.2
Calcium	9.0	9.0	9.5
Phosphorus	6.8	6.8	6.5
Cellulose	65.9	65.9	43.0
NDF	13.89	13.89	10.86
Calculated Composition (g·kg^−1^ as feed bases)			
^†^ ME (MJ/kg)	12.9	12.9	13.7
^†^ NE (MJ/kg)	9.8	9.8	9.97
^†^ SID Lysine	6.3	6.3	7.8
SID Met + Cys	4.9	4.9	4.9
Analysed Fatty Acids Composition (% total FAME)			
ALA, 18:3 n-3	2.40	5.07	4.30
LA, 18:2 n-6	49.60	48.12	48.87
⅀ SAT	15.76	16.66	16.93
⅀ MUFA	33.05	29.89	28.63
⅀ PUFA	52.34	53.46	54.11
⅀ n-3 PUFA	2.64	5.27	4.81
⅀ n-6 PUFA	49.70	48.19	49.30
n-6:n-3 ratio	18.82	9.14	10.24

Abbreviations: lactating sows diets based on soybean oil (SO) and based on hemp seed oil (HO); suckling piglets’ diet (PSO/PHO); FAME = fatty acid methyl esters; ALA = alpha linolenic acid; LA = linoleic acid; ⅀ SFA = total saturated fatty acids; ⅀ MUFA = total monounsaturated fatty acids; ⅀ PUFA = total polyunsaturated fatty acids; ⅀ SFA = 14:0 + 16:0 + 18:0 + 20:0; ⅀ MUFA = 16:1 + 18:1; ⅀ n-6 PUFA = 18:2 n-6 + 20:2 n-6 + 20:2 n-6; Total n-3 PUFA = 18:3 n-3 + 18:4 n-3; ⅀ PUFA = ⅀ n-6 PUFA + ⅀ n-3 PUFA. ^†^ ME was calculated based on feed composition and theoretical regression coefficients; ^†^ NE was calculated using the EvaPig tool, version 2.0.3.2 (2020), developed by the French National Institute for Agricultural Research, METEX NØØVISTAGO, and the French Association of Zootechnie; ^†^ SID: standard ileal digestibility values: calculation based on amino acids contents in the feedstuff and theoretical standardised ileal coefficient from CVB Feed, 2021 [27]. Feed was provided to piglets starting 7 days after birth. ^‡^ Vitamin mineral premix supplied per kg feed for sows: 9000 IU vitamin A; 1500 IU vitamin D3; 50 IU vitamin E; 2 mg vitamin K3; 1.5 mg vitamin B1; 5.2 mg vitamin B2; 15 mg vitamin B3; 8.1 mg vitamin B5; 2 mg vitamin B6; 0.10 mg vitamin B7; 0.5 mg vitamin B9; 0.03 mg vitamin B12; 39 mg of Mn; 100 mg of Fe; 15 mg Cu; 100 mg Zn; 0.3 mg I; 0.22 mg Se; 0.25 mg Co; 60 mg antioxidant. ^‡^ Vitamin mineral premix supplied per kg feed for piglets: 10,000 IU vitamin A; 2000 IU vitamin D3; 30 IU vitamin E; 3 mg vitamin K3; 2 mg vitamin B1; 6 mg vitamin B2; 20 mg vitamin B3; 13.5 mg vitamin B5; 3 mg vitamin B6; 0.06 mg vitamin B7; 0.8 mg vitamin B9; 0.05 mg vitamin B12; 10 mg vitamin C; 30 mg of Mn; 110 mg of Fe; 25 mg Cu; 100 mg Zn; 0.38 mg I; 0.36 mg Se; 0.3 mg Co; 60 mg antioxidant.

**Table 2 vetsci-09-00352-t002:** Centesimal essential fatty acids composition of hemp seed oil compared to soybean oil.

Fatty Acids (% Total FAME)	Soybean Oil	Hemp Seed Oil
ALA, 18:3 n-3	8.05	14.78
LA, 18:2 n-6	55.29	54.12
⅀ SAT	14.30	13.54
⅀ MUFA	22.09	16.62
⅀ PUFA	63.36	69.62
⅀ n-3 PUFA	8.05	14.78
⅀ n-6 PUFA	55.29	54.12
n-6:n-3 ratio	6.86	3.67

Abbreviations: FAME = fatty acid methyl esters; ALA = alpha linolenic acid; LA = linoleic acid; ⅀ SFA = total saturated fatty acids; ⅀ MUFA = total monounsaturated fatty acids; ⅀ PUFA = total polyunsaturated fatty acids; ⅀ SFA = 14:0 + 16:0 + 18:0 + 20:0; ⅀ MUFA = 16:1 + 18:1; ⅀ n-6 PUFA = 18:2 n-6 + 20:2 n-6 + 20:2 n-6; ⅀ n-3 PUFA = 18:3 n-3 + 18:4 n-3; ⅀ PUFA = ⅀ n-6 PUFA + ⅀ n-3 PUFA + CLA.

**Table 3 vetsci-09-00352-t003:** Chemical composition of milk from sows fed a classical diet based on soybean oil (SO) or a rich n-3 FA and a lower n-6:n-3 ratio diet based on hemp seed oil (HO) at different time points after farrowing (12 h, 10 days, and 21 days).

Items	Chemical Composition %				
	Total Solids	Fat	Protein	Lactose	pH
Between-Subjects Factors-Sow diet					
SO	24.11	8.11	6.24	5.22	6.47
HO	24.00	7.95	6.67	5.05	6.20
Within-Subjects Factors-Time					
T1	27.43 ^a^	8.92	11.0 ^a^	3.94 ^a^	6.22
T2	25.00 ^a^	7.85	4.64 ^b^	5.42 ^b^	6.43
T3	19.71 ^b^	7.28	3.81 ^b^	6.00 ^b^	6.30
SEM	0.79	0.42	0.80	0.21	0.06
*p*-Value					
Diet	NS	NS	NS	NS	NS
Time	***	NS	***	***	NS
Interaction	NS	NS	NS	NS	NS

Abbreviations: standard error of mean, SEM; T1 = 12 h after farrowing, T2 = 10 days, T3 = 21 days. *** *p* ≤ 0.001 highly significant difference between means; ^a, b^ Different superscript letters indicate significantly different means (*p* < 0.05).

**Table 4 vetsci-09-00352-t004:** FAs composition of milk from sows fed a classical diet based on soybean oil (SO) or a rich n-3 FA and lower n-6:n-3 ratio diet based on hemp seed oil (HO) at different time points after farrowing (12 h, 10 days, and 21 days).

Items	Fatty Acids Composition (% of Total FAME)							
	⅀ SAT	⅀ MUFA	⅀ PUFA	⅀ n-3	⅀ n-6	ALA	LA	n-6/n-3
Between-Subjects Factors-Sow diet								
SO	31.42	43.48 ^a^	25.00 ^a^	0.97 ^a^	24.06 ^a^	0.51 ^a^	23.40 ^a^	28.30 ^a^
HO	32.08	35.70 ^b^	31.47 ^b^	1.68 ^b^	29.59 ^b^	1.61 ^b^	28.64 ^b^	17.61 ^b^
Within-Subjects Factors -Time								
T1	27.30 ^a^	36.27 ^a^	35.00 ^a^	1.64 ^a^	33.70 ^a^	1.43 ^a^	33.6 ^a^	20.54 ^a^
T2	31.63 ^b^	46.42 ^b^	20.59 ^b^	1.04 ^a^	18.98 ^b^	0.83 ^b^	18.4 ^b^	18.25 ^b^
T3	35.19 ^c^	43.30 ^c^	21.39 ^c^	1.32 ^a^	19.87 ^b^	1.22 ^b^	19.0 ^c^	15.08 ^c^
SEM	3.61	4.28	4.59	0.24	0.68	0.30	3.68	2.83
*p*-Value								
Diet	NS	**	*	*	*	**	***	**
Time	***	***	***	**	***	**	***	***
Interaction	NS	***	*	*	***	*	***	*

Abbreviations: standard error of mean, SEM; fatty acids methyl ester, FAME; ⅀ SAT= total saturated fatty acids;.⅀ SAT= C4:0 + C6:0 + C8:0 + C10:0 + C12:0 + C14:0 + C15:0 + C16:0 + C17:0 + C18:0 + C20:0 + C23:0; ⅀ MUFA = monounsaturated fatty acids; ⅀ MUFA = C14:1 + C15:1 + C16:1 + C17:1 + C18:1n-9c + C18:1n-7c + C22:1n-9; ⅀ PUFA = C18:2n-6 + C18:3n-3 + CLA + C18:4n-3 + C20:2n-6 + C20:3n-6 + C20:3n-3 + C20:4n-6 + C22:2n-6 + C20:5n-3 + C22:4n-6 + C22:6n-3; ⅀ n-3= C18:3n-3 + C18:4n-3 + C20:3n-3 + C20:5n-3 + C22:6n-3; ⅀ n-6 = C18:2n-6 + C20:2n-6 + C20:3n-6 + C20:4n-6 + C22:2n-6 + C22:4n-6; total PUFA = total n-6 PUFA + total n-3 PUFA + total CLA; ALA = α-linolenic FA; LA= linoleic FA; CLA = conjugated linoleic acid; T1 = 12h after farrowing, T2 = 10 days, T3 = 21 days; * *p* < 0.05 significant difference between means, ** *p* < 0.01 and *** *p* ≤ 0.001 highly significant difference between means; ^a, b, c^ Different superscript letters indicate significantly different means (*p* < 0.05).

**Table 5 vetsci-09-00352-t005:** Ig composition of milk from sows fed a classical diet based on soybean oil (SO) or a rich n-3 FA and lower n-6:n-3 ratio diet based on hemp seed oil (HO) at different time points after farrowing (12 h, 10 days, and 21 days).

		Milk Igs	
Ig (mg/mL)	IgA	IgG	IgM
Between-Subjects Factors-Sow diet			
SO	4.10	17.43	2.11 ^a^
HO	4.69	19.27	3.54 ^b^
Within-Subjects Factors-Time			
T1	9.24 ^a^	48.33 ^a^	4.63 ^a^
T2	1.81 ^b^	4.15 ^b^	1.29 ^b^
T3	2.17 ^b^	2.88 ^b^	2.90 ^b^
SEM	0.69	4.17	0.34
*p*-Value			
Diet	NS	NS	***
Time	***	***	***
Interaction	NS	NS	*

Abbreviations: standard error of the mean, SEM. T1 = 12 h after farrowing, T2 = 10 days, T3 = 21 days; NS: nonsignificant effect; * *p* < 0.05 significant difference between means; *** *p* ≤ 0.001 highly significant difference between means; ^a, b^ Different superscript letters indicate significantly different means (*p* < 0.05).

**Table 6 vetsci-09-00352-t006:** Ig composition of plasma from sows fed a classical diet based on soybean oil (SO) or a rich n-3 FA and lower n-6:n-3 ratio diet based on hemp seed oil (HO) at different time points after farrowing (12 h, 10 days, and 21 days).

		Sow Plasma	
Ig (mg/mL)	IgA	IgG	IgM
Between-Subjects Factors-Sow diet			
SO	2.41	17.33	6.38
HO	2.38	18.45	6.34
Within-Subjects Factors-Time			
T1	8.09 ^a^	17.99 ^a^	5.68 ^a^
T2	2.23 ^b^	16.95 ^a^	5.93 ^b^
T3	3.30 ^c^	18.93 ^a^	7.45 ^c^
SEM	0.15	2.15	1.04
*p*-Value			
Diet	NS	NS	NS
Time	***	NS	***
Interaction	NS	NS	NS

Abbreviations: standard error of the mean, SEM; T1 = 12 h after farrowing; T2 = 10 days; T3 = 21 days; NS: nonsignificant effect; *** *p* ≤ 0.001 highly significant difference between means. ^a, b, c^ Different superscript letters indicate significantly different means (*p* < 0.05).

**Table 7 vetsci-09-00352-t007:** Spearman coefficient correlation between Ig of plasma and milk from sows and FA.

Item			Milk							
			SFA	MUFA	PUFA	n-3	n-6	n-6:n-3	ALA	LA
**Milk**	IgA	r	−0.58	−0.70	0.65	0.23	0.58	0.54	−0.19	0.61
		*p*-value	**	***	***	NS	**	**	NS	**
	IgG	r	−0.88	−0.40	0.65	0.15	0.58	0.59	−0.27	0.60
		*p*-value	***	NS	***	NS	**	**	NS	**
	IgM	r	−0.45	−0.63	0.54	0.13	0.45	0.56	−0.14	0.50
		*p*-value	NS	***	**	NS	NS	**	NS	**
**Plasma**	IgA	r	0.81	0.43	−0.75	0.08	−0.68	−0.67	0.24	−0.71
		*p*-value	***	NS	***	NS	***	***	NS	***
	IgG	r	0.01	−0.21	0.01	0.42	−0.01	0.04	−0.05	0.06
		*p*-value	NS	NS	NS	NS	NS	NS	NS	NS
	IgM	r	−0.71	0.19	−0.59	0.16	−0.56	−0.47	0.10	−0.52
		*p*-value	***	NS	**	NS	**	NS	NS	**

NS: nonsignificant effect; ** *p* < 0.01 and *** *p* ≤ 0.001 highly significant difference between means.

**Table 8 vetsci-09-00352-t008:** Ig composition of plasma from piglets from sows fed a classical diet based on soybean oil (PSO) or a rich n-3 FA and lower n-6:n-3 ratio diet based on hemp seed oil (PHO) at different time points after farrowing (12 h, 10 days, and 21 days).

Igs (mg/mL)	Piglets Plasma		
	IgA	IgG	IgM
Between-Subjects Factors-Sow diet			
PSO	1.62	26.22	2.04
PHO	1.74	27.35	2.10
Within-Subjects Factors-Time			
T1	2.83 ^a^	43.12 ^a^	2.88 ^a^
T2	1.54 ^b^	25.78 ^b^	1.41 ^b^
T3	0.71 ^c^	11.64 ^c^	1.93 ^b^
SEM	0.18	2.61	0.17
*p*-Value			
Sow diet	NS	NS	NS
Time	***	***	**
Interaction	NS	NS	NS

Abbreviations: standard error of the mean, SEM. NS: nonsignificant effect; ** *p* < 0.01 and *** *p* ≤ 0.001 highly significant difference between means. T1 = 12 h after farrowing, T2 = 10 days, T3 = 21 days. ^a, b, c^ Different superscript letters indicate significantly different means (*p* < 0.05).

**Table 9 vetsci-09-00352-t009:** Spearman coefficient correlation in piglets.

Item			Milk							
			⅀ SFA	⅀ MUFA	⅀ PUFA	⅀ n-3	⅀ n-6	n-6:n-3	ALA	LA
**Piglets** **Plasma**	IgA	r	−0.88	−0.42	0.73	0.14	0.69	0.63	−0.36	0.67
		*p*-value	***	NS	***	NS	***	***	NS	***
	IgG	r	−0.87	−0.42	0.66	0.07	0.64	0.57	−0.17	0.63
		*p*-value	***	NS	***	NS	***	**	NS	***
	IgM	r	−0.52	−0.57	0.54	0.09	0.54	0.55	−0.09	0.60
		*p*-value	**	**	**	NS	**	**	NS	**

NS: nonsignificant effect; ** *p* < 0.01 and *** *p* ≤ 0.001 highly significant difference between means.

**Table 10 vetsci-09-00352-t010:** E-CH_4_ production of sows fed a classical diet based on soybean oil (SO) or a rich n-3 FA and lower n-6:n-3 ratio diet based on hemp seed oil (HO) at different time points after farrowing (12 h, 10 days, and 21 days).

Items	E-CH_4_ (g CO_2_ eq·day^−1^)					
	Sow		Litter		Total	
	SO	HO	PSO	PHO	SO (PSO)	HO (PHO)
Sows diet	264	252	11.3	8.4	276	261
Period						
T1	202 ^a^	188 ^a^			202 ^a^	188 ^a^
T2	334 ^c^	297 ^c^			334 ^c^	297 ^c^
T3	271 ^b^	257 ^b^	7.9	7. 8	279 ^b^	265 ^b^
SEM	18.11	16.14	1.60	0.94	18.06	16.36
*p*-Value						
Sow diet	NS		NS		NS	
Period	***		*		***	
Interaction	NS		NS		NS	
**Model**			**β Coefficient**	** *R* **	***R*-Square**	***p*-Value**
1				0.99	0.99	
Constant			8.0			
Fat intake			1.17			***
2				0.69	0.46	
Constant			−117			
n-6:n-3 PUFA			−3.15			***
3				0.54	0.28	
Constant			244			
ALA			107			**
4				0.45	0.20	
Constant			−516			
Lean meat, %			13.6			*

Abbreviations: standard error of the mean, SEM. We took into account the global warming potential of 25 for CH_4_. ^a^ Dependent variable, E-CH_4_ and its potential predictors. T1 = 12 h after farrowing, T2 = 10 days, T3 = 21 days; NS: nonsignificant effect; * *p* < 0.05 (significant difference between means); ** *p* < 0.01 and *** *p* ≤ 0.001 (highly significant difference between means). ^a, b, c^ Different superscript letters indicate significantly different means (*p* < 0.05).

## Data Availability

Not applicable.

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
