# Peer review of "Alterations in Essential Fatty Acids, Immunoglobulins (IgA, IgG, and IgM), and Enteric Methane Emission in Primiparous Sows Fed Hemp Seed Oil and Their Offspring Response"

_vetsci, 2022, doi:10.3390/vetsci9070352_

Round 1

Reviewer 1 Report

The author did well in response to reviewer comment and try to revised manuscript. The objective and result explanation are more clear, statistically analysis was reconsider, as well as discussion. Interaction between main effects was mentioned although not clear all but acceptable. Mistake data were rechecking and edited. Only one point has to recheck which is intake of sow that presented in previous version manuscript but not present version. If the author decided to not present or emphasize, suitable reasons have to clarify. Some comments are in the attached file

Author Response

General comments

Thank you for RW appreciations.

We accepted and reviewed the manuscript according to the suggestion of the RW.

Specific comment

REVIEWER.

Result

RW: no results on intake of sows as presented in the previous manuscript….

Response 1. Line 347 we added the following paragraph:

Primiparous sows' feed, fat, DM, and ME intake increased linearly until 21 days (weaning period, + about 40%, being 3.56 on average the first days and rising to 5.96 at weaning, P<0.0001), irrespective of diet (data presented in supplementary Table 1S).

Data on sow intake is shown in supplementary Table 1S in order to simplify the Results chapter. Even more, the diet had no significant influence on intake.

  1. RW. Figure 1a and 1b , body weight (BW).

Response 2. We corrected the abbreviations for body weight and metabolic body weight in Figure 1a and 1b.

Reviewer 2 Report

The subject matter of the authors is very interesting in several respects. However, some adjustments can be made to improve the article. In particular, simplify the results, which are too articulate, and expand the discussion on CH4 emissions. Attention to bibliographical references and editing

Author Response

General comments

We accepted and reviewed the manuscript according to the suggestion of the RW.

Specific comment

Abstract

  1. Line 17-19… n-6 :n-3 polyunsaturated fatty acids (PUFA)

Response 1. Line 17-19: we corrected the style for n-6 :n-3 polyunsaturated fatty acids (PUFA)

Introduction.

  1. RW.Attention to bibliographical references and editing. Line 77 and 90. Correct the reference numbers

Response 2. We checked all the references throughout the manuscript. Line 77. We replaced FAO [17] with FAO [15].

                   Line 90. We changed the number into the Reference, respectively 27 and 28 become 25 and 26:

  1. Lauridsen, C., and V.  Danielsen. Lactational dietary fat levels and sources influence milk composition and performance of sows and their progeny. Prod. Sci. 2004, 91:95–105. doi:10.1016/j.livprodsci.2004.07.014.
  2. Lauridsen, C. Effects of dietary fatty acids on gut health and function of pigs pre- and post-weaning. J. Anim. Sci. 98, 4, 1–12. doi:10.1093/jas/skaa08.

  1. Line 85-88. RW suggested to move the paragraph above (line 48-50)

Response 3. Line 85-88. We moved the next paragraph “In previous studies aimed to increase litter performance and positively influenced their health, specific attention was given to the colostrum and milk FA composition as well as the biological responses of offspring” to line 48-50.

Material and Methods

  1. Clarify NE (Table 1)

Response 4. We added below Table 1: NE was calculated using EvaPig tool, version 2.0.3.2 (2020), developed by the French National Institute for Agricultural Research, METEX NØØVISTAGO, and the French Association of Zootechnie.

Results

  1. RW. In particular, simplify the results, which are too articulate, and expand the discussion on CH4 emissions. 

Response 5. We deleted all the sentences where no significant differences were recorded:

# Line 370-371. Milk chemical composition.

As shown in Table 3, the diet did not affect the chemical milk components. On the contrary,…

# Line 395-396. The sentence “The effect of time point was more pronounced for PUFA, ⅀n-6, and LA than ALA and ⅀n-3” was moved to line 648 in the Discussion Chapter.

  #  Line 424-426. The sentence “during the entire lactation period the Ig average values were: 4.40 mg/mL IgA, 18.45 mg/mL IgG and 2.94 mg/mL IgM “was moved to the Discussion Chapter line 662-663.

# Line 333. We deleted the sentence “As shown in Table 6, the dietary hemp seed oil did not change the Igs concentration on plasma from sows (p > 0.05)” due to the fact that the diet did not affect significantly the Ig concentration.

# Line 483-486.  We moved the sentence “Piglets were fed only by maternal secretion for the first 10 days after birth, after which their diet was supplemented with a basal solid compound feed. The variation in the variables assayed could be attributed to their sows-mother feed composition. Surprisingly, the sows’ diets did not affect the litter Igs concentration in plasma, although a slight increase was observed regardless of the kind of Ig” to Discussion Chapter line 679-680.

#Line 534-536. We moved the sentence “Suckling piglets should not produce CH4 since sow's milk does not contain fibre or polysaccharides. On the other hand, the solid feed contains fibre, which results in low CH4 emissions” to the Discussion Chapter line 742-743.

  1. Line 444-445. Delete the sentence: There were no significant interactions between diets and period regarding Igs concentration on plasma.

Response 6. Line 444-445. As suggested by the RW the sentence was deleted.

  1. RW. Table 7. Attention to editing

Response 7 We reediting Table 7.

Discussion

  1. Expand the discussion on CH4 emissions

Response 8. As suggested the RW we added new comments and references related to CH4:

Line 734-738 “Compared to other productive animals, like cattle, the CH4 emissions from pigs represent a very small percentage of all generated emissions. There are some investigations which showed the inhibitory effect of ALA to the CH4 in ruminants [65]. In contrast to SFA, which are less effective at reducing CH4 in cattle, C12:0, C18:3, and PUFA were mentioned by [65] as providing a considerable influence on decreasing CH4 emission in ruminant.”

References:

  1. Patra, A.K. The effect of dietary fats on methane emissions, and its other effects on digestibility, rumen fermentation and lactation performance in cattle: A meta-analysis. Livestock Sci. 2013, 155(2-3), 244–254. doi:10.1016/j.livsci.2013.05.023 

Line 743-747 “Our findings are in accordance with those of Nguyen et al., 2020 [66], who reported that using different n-6 to n-3 FAs ratios in diets based on corn-soybean meal seemed to have no effect on gas emission in growing pigs. We presume that the type of diet and the concentration of EFAs may influence how much E-CH4 is reduced. On the other hand, suckling piglets should not produce CH4 since sow's milk does not contain fibre or polysaccharides. The solid feed contains fibre, which results in low CH4 emissions.

References:

  1. Nguyen, D.H.; Yun, H.M and Kim, I.H. Evaluating Impacts of Different Omega-6 to Omega-3 Fatty Acid Ratios in Corn–Soybean Meal-Based Diet on Growth Performance, Nutrient Digestibility, Blood Profiles, Fecal Microbial, and Gas Emission in Growing Pigs. Animals 2020, 10, 42; doi:10.3390/ani10010042.

This manuscript is a resubmission of an earlier submission. The following is a list of the peer review reports and author responses from that submission.

Round 1

Reviewer 1 Report

Alterations in Essential Fatty Acids, Immunoglobulins (IgA, IgG and IgM), and Enteric Methane Emission in Primiparous Sows Fed Hemp Seed Oil and Their Offspring Response

Mihaela Habeanu and co-authors performed an interesting study aiming to evaluate the effects of different oil sources (soybean and hemp) on essential fatty acids, immunoglobulins, and enteric methane emission. However, some points need to be reviewed carefully, before the study can be considered for publication.

Abstract

Lines 14-16: the sentences are too long, please changed it.

Lines 19 and 20: Did you “noticed”? What do you mean? Did you obtain results or not? Please clarify

Introduction

Lines 29 and 30: Please change linoleate and linoenate with linoleic acids and alpha-linolenic acids.

Line 31: please control the layout of references number

Line 48-54: maybe could be better add a reference in this period.

Line 55: please change proposed with suggested

Line 81-82: Please add the following article: https://doi.org/10.1016/j.indcrop.2018.12.084

Material and methods

Line 133: please delete “depending on the type of oil used”

Line 146: “ME and was calculated …” ME and what? Please correct

Lines 161-163: Please clarify these sentences. What do mean with multiple simulations?

Line 203: please delate “packages”

Line 209: please correct this sentence

Line 214: I have some doubt about the analysis of THC, the reference described in the text doesn’t report the analysis specifically, can you give more details?

Statistical analysis: I have some doubt regarding the experimental scheme. The number of pens is low, maybe would be better perform a non-parametric-test regarding the results obtained in sows.

Results

Line 311-312: “predominance of ALA”, did you observe significant results?

Line 315: regarding the analysis of THC, maybe would be better analyse the CBD content

Line 327: “The initial BW was similar between groups” please delate

Line 327-328: Please delate this sentence, the results were not significant

Table 3: Please control the layout for period effect. Moreover, it’s better write “interection” rather than p-Value. Please reported the mean of “a”, “b”, and “c”.

Line 323-333 Any comment about the interaction of meet quality?

Line 340: Please delate “(distinctly significant difference between means)

Line 344: the results are not significant (p>0.05), please avoid reporting not significant results.

Line 365: You already reported “Table 5”at beginning of period, In my opinion it’s not necessary to repeat.

Lines: 368-375: Did you comment the results obtained by diet and period effects. However, could be interesting reported the results about interaction.

Table 5: attention to the number after the point, two o one? Please delate T, you cannot report results which are not significant. Indicate what a, b, and c mean. n6/n3 ratio, in some cases you reported the highest value with a in other case with b. Please decide. Particularly for period effect, sound strange that some results are not significant. For example, Lactose in period effect.

Line 389: again, delate sentences about no significant differences

Line 396: “tendency” what do you mean? is it significant? If it is not significant, please delate

Table 6: IgM, 2.11 and 3.54 not significant, also, 17.43 and 19.27. Sound strange. Please define the number after the point. P-value, change with interaction.

Lines 424-433: maybe could be better reported the correlation results in table or figure. Reporting the number of replication and p-value.

Line 465: Please reported the p-value

Discussion

In general, would be better improve all the discussion paragraph

Maybe in this section should be better justify the stillborn during the trial.

  • Lines 503-505: Please add the following studies: doi.org/10.3390/ani11030856; doi.org/10.1080/1828051X.2022.2039562

Lines 532-536: did you compare your results with other authors, which tested diets with different tryptophan:lysine ratio. However, you didn’t report this results in your diets.

Reviewer 2 Report

The manuscript Vetsci-1730909

reports date from a study that evaluated offspring responses after different oil supplementation to sows. The manuscript is not well organized, lacks relevant information, and makes readers hard to follow what the authors want to convey.

  1. Please use the full word with the abbreviation in the Abstract (SO, HO, etc)
  2. L19-20: include the data and statistics
  3. No objective nor conclusion was stated in the abstract
  4. L36-40: This does not support the rationale of this study. Suggest deleting the sentence.
  5. L40-42: Didn’t get the point. Please re-write.
  6. L42-48: Summarize and shorten the sentences
  7. L62-63: What’s the connection between reducing antibiotics and increasing attention to reducing greenhouse gases?
  8. L68: Monogastric animal
  9. L70-80: Need to be summarized
  10. L82-85: Hard to understand
  11. L94-95: Remove the sentence
  12. Overall, the introduction section is too crowded, not well summarized, and hard to understand what the authors want to emphasize.
  13. L133: Is SO is control sow diet?
  14. L157-158: Re-write the sentence
  15. L161-171: Remove replicated information
  16. L177: 10 days
  17. L206-207: reference needed
  18. L240-242: remove replicated information
  19. L257-258: Frequency = [(total number of piglets with diarrhoea x days of diarrhoea)/(total number of piglets x days of the experiment)] x 100%
  20. L327: Check the style
  21. L332: R or R^2 needed
  22. L333: Data need to be stated
  23. Table 3: Superscripts need to be removed if there’s no differences
  24. Table 3: Check the style and alignment
  25. L344-345: Difficult to understand
  26. Table 4: Suggest to move to the supplementary data
  27. L355-500: Check the table style, size, alignment, statistical differences; There’s no uniformity in the result section, it is hard to follow.
  28. L501: summarize the results/findings in this experiment
  29. L508-512: Not appropriate in the discussion section
  30. L520-521: Too broad. Please include more specific information to support
  31. L522-526: this can’t be the single paragraph
  32. L532-536: so what’s the potential mechanisms of this finding?
  33. L537-541: It is difficult to understand why this sentence was written
  34. L542-645: Authors didn’t discuss ‘how’ and ‘why’ they observed the results. The statement was just based on the results and should be discussed based on the references.
  35. Overall, the discussion was not enough, hard to follow what the authors wanted to convey, and not appropriate ways of discussion.

Reviewer 3 Report

Immune response of sucking piglets when their sow received different PUFA ratio in feed was investigated by using standard method and laboratory techniques. The result did clear for main objective including estimated CH4 emission which is another point of this work. Discussion of finding result with previous work with relates reason did great also.

However, I have big wonder about treatment arrangement that set period as main effect and also statistical analysis and presentation of the result. No need to compare time after farrowing due to that data come from 2 PUFA ratios, it doesn’t explain the main effect which is SO vs HO as present in Introduction. Several parameters were found significant in DietxPeriod interaction but the author did not mind. Moreover, some data may be mistake such as in Table 3, Table 9, value of 12h-21d period should be between value of 12h-10d and 10d-21d. Therefore, this manuscript should reconsider by statistically expertise.

I put some more comments in the attached file
